

# Analysing surface energy balance closure and partitioning over a semi-arid savanna FLUXNET site in Skukuza, Kruger National Park, South Africa

Nobuhle P. Majozi[1,2], Chris M. Mannaerts[2], Abel Ramoelo[1], Renaud Mathieu[1,3], Alecia Nickless[4], Wouter Verhoef[2]

[1]Earth Observation Group, Natural Resources and Environment, Council for Scientific and Industrial Research, Pretoria, South Africa, 0001
[2]Department of Water Resources, Faculty of Geo-Information Science and Earth Observation (ITC), University of Twente, Enschede, 75AA, the Netherlands
[3]Department of Geography, Geoinformatics and Meteorology, University of Pretoria, South Africa
[4]Nuffield Department of Primary Care Health Sciences, University of Oxford, Oxford, OX2 6GG, United Kingdom

*Correspondence to*: N. P. Majozi (nmajozi@csir.co.za)

**Abstract:** Flux tower data are in high demand to provide essential terrestrial climate, water and radiation budget information needed for environmental monitoring and evaluation of climate change impacts on ecosystems and society in general. They are also intended for calibration and validation of satellite-based earth observation and monitoring efforts, such as for example assessment of evapotranspiration from land and vegetation surfaces using surface energy balance approaches.

Surface energy budget methods for ET estimation rely to a large extend on the basic assumption of a surface energy balance closure, assuming the full conversion of net solar radiation reaching the land surface into soil heat conduction and turbulent fluxes, i.e. the sensible (or convection) and latent heat components of the energy balance.

In this paper, the Skukuza flux tower data were analysed in order to verify their use for validation of satellite–based evapotranspiration methods, under development in South Africa. Data series from 2000 until 2014 were used in the analysis. The energy balance ratio (EBR) concept, defined as the ratio between the sum of the turbulent convective and latent heat fluxes and radiation minus soil heat was used. Then typical diurnal patterns of EB partitioning were derived for four different seasons, well illustrating how this savannah-type biome responds to weather conditions. Also the particular behaviour of the EB components during sunrise and sunset conditions, being important but usually neglected periods of energy transitions and inversions were noted and analysed.
Annual estimates and time series of the surface energy balance and its components were generated, including an evaluation of the balance closure. The seasonal variations were also investigated as well as the impact of nocturnal observations on the overall EB behaviour.

## 1    Introduction

The net solar radiation (Rn) reaching the earth's surface determines the amount of energy available for

transformation into energy balance components, i.e. latent (LE), sensible (H) and ground (G) heat fluxes,

including heat stored by the canopy and the ground. Energy partitioning on the earth's surface is a function of

interactions between biogeochemical cycling, plant physiology, the state of the atmospheric boundary layer and

climate (Wilson et al., 2002).  How the turbulent fluxes (sensible and latent heat fluxes) are partitioned in an

ecosystem plays a critical role in determining the hydrological cycle, boundary layer development, weather and

climate (Falge et al., 2005). Understanding the partitioning of energy, particularly the turbulent fluxes, is

important for water resource management in (semi) arid regions, where potential evapotranspiration far exceeds

precipitation.

Eddy covariance (EC) systems are currently the most reliable method for measuring carbon, energy and

water fluxes, and they have become a standard technique in the study of surface-atmosphere boundary layer

interactions. Hence, they provide a distinct contribution to the study of environmental, biological and





climatological controls of the net surface exchanges between the land surface (including vegetation) and the
atmosphere  (Aubinet, et al., 1999; Baldocchi et al., 2001). The accuracy of these data is very important because
they are used to validate and assess performance of land surface and climate models. However, the eddy
covariance techniques have limitations in terms of data processing and quality control methods, especially under
complex conditions (e.g., unfavorable weather, such as high turbulence and low wind speed, and heterogeneous
topography). In EC measurements, the ideal situation is that available energy, i.e. net radiation minus soil heat
flux is equal to the sum of the turbulent fluxes (latent and sensible heat fluxes) (Rn-G = LE+H); however, in
most instances, the available energy (i.e. net radiation–soil heat flux (Rn-G)) is larger than the sum of the
measurable turbulent fluxes of sensible heat and latent heat. Extensive research investigated and reported the
issue  of surface energy imbalance in EC observations (Barr et al., 2012; Chen et al., 2009; Foken et al., 2010;
Franssen et al., 2010; Mauder et al., 2007), and this closure error (or imbalance) has been documented to be
around 10-30 %. Causes for non-closure include  unaccounted soil and canopy heat storage, non-inclusion of the
low frequency turbulence in the computation of the turbulent fluxes, land surface heterogeneities, systematic
measurement and sampling errors. This imbalance has implications on how energy flux measurements should be
interpreted and how these estimates should be compared with model simulations. The surface energy balance
closure is an accepted validation procedure of eddy covariance data quality (Twine et al., 2000; Wilson et al.,
2002), and different methods have been used to assess the energy closure and partitioning, including ordinary
least squares regression (OLS) method, the residual method, i.e. Residual = Rn-G-H-LE, and the energy balance
ratio, i.e. EBR = LE+H/Rn-G.

69        Several researchers have investigated surface energy partitioning and energy balance closure for

different ecosystems, including savannas. Bagayoko et al. (2007) examined the seasonal variation of the energy
balance in West African savannas, and noted that latent heat flux played a major role in the wet season, whereas
sensible heat flux was significant in the dry season. In the grassland Mongolian Plateau, Li et al. (2006)
concluded that sensible heat flux dominated the energy partitioning, followed by ground heat flux, even during
the rainy season, which showed a slight increase in latent heat flux. Gu et al. (2006) used different ratios
(Bowen ratio, G/Rn, H/Rn and LE/Rn) to investigate surface energy exchange in the Tibetan Plateau, and
showed that during the vegetation growth period, LE was higher than H, and this was reversed during the post-
growth period.

78        Research on the South African savanna, i.e. using data from the Skukuza EC system, has focused mainly on

the carbon exchange, fire regimes, and in global analysis of the energy balance (Archibald et al., 2009; Kutsch
et al., 2008; Williams et al., 2009). Hence, the need to explore the surface energy partitioning and energy
balance closure of this ecosystem. In this study, we will examine the surface energy balance partitioning into
soil heat conduction, convection (sensible) and latent heat components and its energy balance closure using 15
years (2000-2014) of eddy covariance data from the Skukuza flux station.

84        First, a multi-year surface energy balance closure analysis was done, and annual values for the EB

components derived, including an evaluation of the balance closure and its error sources. To further investigate
the EB partitioning and closure at this location and biome, the seasonal effect on the EB variations was also
assessed. Thirdly, the effect of nocturnal (nighttime) observations on the overall daily EBR was verified. Then
the partitioning of the net solar radiation into soil heat and the turbulent fluxes (L+HE) during the different



seasons was assessed at the sub-daily (30-min) scale for this African savanna system for the year 2012 with
meteorological data.

**2    Materials and methods**
**2.1    Site description**
The Skukuza flux tower (25.02°S, 31.50°E) was established early 2000 as part of the SAFARI 2000 campaign
and experiment, set up to understand the interactions between the atmosphere and the land surface in southern
Africa by connecting ground data of carbon, water, and energy fluxes with remote sensing data generated by
Earth observing satellites (Scholes et al., 2001; Shugart et al., 2004).

98        The site is located in the Kruger National Park (South Africa) at 365 m above sea level, and receives

550 ± 160 mm precipitation per annum between November and April, with significant inter-annual variability.
The year is divided into a hot, wet growing season and a warm, dry non-growing season. The soils are generally
shallow, with coarse sandy to sandy loam textures (about 65 % sand, 30 % clay and 5% silt).  The area is
characterised by a catenal pattern of soils and vegetation, with broad-leaved *Combretum* savanna on the crests
dominated by the small trees (*Combretum apiculatum*), and fine-leaved *Acacia* savanna in the valleys dominated
by *Acacia nigrescens* (Scholes et al., 1999). The vegetation is mainly open woodland, with approximately 30 %
tree canopy cover of mixed *Acacia* and *Combretum* savanna types. Tree canopy height is 5–8 m with occasional
trees (mostly *Sclerocarya birrea*) reaching 10 m. The grassy and herbaceous understory comprises grasses such
as *Panicum maximum*, *Digitaria eriantha*, *Eragrostis rigidor*, and *Pogonarthria squarrosa*.

**2.1.1    Eddy covariance system**
Since 2000, ecosystem-level fluxes of water, heat and carbon dioxide are measured using an eddy covariance
system mounted at 17 m height of the 22 m high flux tower. The measurements taken and the instruments used
are summarised in Table 1.

**(Table 1)**

From 2000 to 2005, H and LE were derived from a closed-path $CO_2/H_2O$ monitoring system, which was
replaced by the open-path gas analyser in 2006. Also, from 2000 to 2008, incident and reflected shortwave
radiation (i.e. 300–1100 nm, $Wm^{-2}$), incident and reflected near-infrared (600–1100 nm, $Wm^{-2}$) and incoming
and emitted longwave radiation (>3.0 µm, $Wm^{-2}$) measurements were made using a two-component net
radiometer (Model CNR 2: Kipp & Zonen, Delft, The Netherlands) at 20 s intervals and then recorded in the
data-logger as 30 min averages; this was replaced with the Kipp & Zonen NRlite net radiometer in 2009.

122       Ancillary meteorological measurements include air temperature and relative humidity, also measured at

16 m height, using a Campbell Scientific HMP50 probe; precipitation at the top of the tower using a Texas
TR525M tipping bucket rain gauge; wind speed and direction using a Climatronics Wind Sensor; and soil
temperature using Campbell Scientific 107 soil temperature probe.





### 2.1.2 Data pre-processing

Post-processing of the raw high frequency (10 Hz) data for calculation of half-hour periods of the turbulent
fluxes of sensible heat (H; W m$^{-2}$), water vapor (LE; W m$^{-2}$), and $CO_2$ ($F_c$; g $CO_2$ m$^{-2}$ time$^{-1}$) involved standard
spike filtering, planar rotation of velocities and lag correction to $CO_2$ and q (Aubinet et al., 1999; Wilczak et al.,
2001). All fluxes are reported as positive upward from the land to the atmosphere. Frequency response
correction of some of the energy lost due to instrument separation, tube attenuation, and gas analyzer response
for LE and $F_c$ was performed with empirical cospectral adjustment to match the H co-spectrum (Eugster and
Senn, 1995; Su et al., 2004).

### 2.2 Data analysis

Half-hourly measurements of eddy covariance and climatological data from 2000 to 2014 were used to assess
surface energy partitioning and closure. Screening of the half-hourly data rejected i) data from periods of sensor
malfunction (i.e. when there was a faulty diagnostic signal), (ii) incomplete 30 minute datasets of net radiation,
ground, latent and sensible heat fluxes, and iii) outliers. After data screening, flux data with non-missing values
of net radiation (Rn), ground heat flux (G), latent heat (LE) flux and sensible heat flux (H) data were arranged
according to monthly and seasonal periods (summer (December – February), autumn (March – May), winter
(June – August), and spring (September – November)), as well as into daytime and nighttime. These data
without gaps were then used to analyse for surface energy balance closure.

### 2.2.1 Surface energy balance assessment


The law of conservation of energy states that energy can neither be created nor destroyed, but is transformed
from one form to another, hence the ideal surface energy balance equation is written as:
$$Rn - G = H + LE \tag{1}$$

Energy imbalance occurs when both sides of the equation do not balance. The energy balance closure was
evaluated for at different levels, i.e. multi-year, seasonal, and day/ night periods,  using two methods, i.e.
i)     The ordinary least squares method (OLS), which is the regression between turbulent fluxes (right side
153        of equation 1, H+LE) and available energy (left side of equation 1, Rn-G)
Ideal closure is when the intercept is zero and slope and the coefficient of determination are one.  This method is
only valid when there are no random errors in the independent variables, i.e Rn and G.
ii)    The energy balance ratio (EBR), i.e. ratio of the sum of turbulent fluxes to the available energy
$$EBR = \frac{\sum (H+LE)}{\sum (Rn-G)} \tag{2}$$

The EBR gives an overall evaluation of energy balance closure at longer time scales by averaging over random
errors in the half-hour measurements; and the ideal closure is 1. EBR has the potential to neglect biases in the
half-hourly data, such as the tendency to overestimate positive fluxes during the day and underestimate negative
fluxes at night.



### 2.2.2 Analysing surface energy partitioning

To evaluate solar radiation variation and partitioning into latent and sensible heat fluxes in this biome, EC surface energy data from 2000 to 2014 were used. The data gaps in these data were first filled using the Amelia II software (Honaker, King, & Blackwell, 2011). This R-program was designed to impute missing data using a bootstrapping-based multiple imputation algorithm. The minimum, maximum and mean statistics of Rn, H, LE and G were then estimated.

To further investigate how meteorological data influence and/or affect the partitioning of the surface energy fluxes, meteorological variables (temperature, precipitation and soil moisture) from 2012 were analysed. The monthly and seasonal variations of energy partitioning were investigated, as well as the energy flux inversions during night-day transitions.

## 3 Results and discussion

### 3.1 Surface energy balance assessment

Data completeness varied largely between 12.76 % (2001) and 57.65 % (2010), with a mean of 36 % and standard deviation 15 %. The variation in data completeness is due to a number of factors including instrument failures, changes and (re)calibration, and poor weather conditions.

### 3.1.1 Multi-year analysis of surface energy balance closure

Fig 1 summarises results of the multi-year energy balance closure analysis for the Skukuza eddy covariance system from 2000 to 2014. The slopes ranged between 0.93 and 1.47, with a mean $1.19 \pm 0.21$, and the intercepts were a mean of $17.79 \pm 32.96$ Wm². $R^2$ ranged between 0.73 in 2005 and 0.92 in 2003, with a mean of 0.86 with standard deviation of 0.05.

The annual energy balance ratio (EBR) ranged between 0.44 in 2007 and 3.76 in 2013, with a mean of $0.97 \pm 0.81$. Between 2004 and 2008, EBR ranges between 0.44 and 0.53, whereas from 2000 to 2003 and 2009 to 2014, the EBR ranged 0.76 and 1.09, with 2013 having an extreme EBR of 3.76. The EBR for 2010 to 2012 were greater than 1, indicating an overestimation of the turbulent fluxes (H+LE) compared to the available energy. The remaining years were less than 1, indicating that the turbulent fluxes were lower than the available energy. The period of low EBR between 2004 and 2008 is characterised by the absence of negative values of available energy (Rn-G, i.e. the nocturnal measurements of fluxes and radiation) as illustrated in Fig 1. Our final proposed mean annual EBR estimate for the (2000-2014) 15-year period, excluding those with data issues (2004 to 2008, and 2013), was therefore 0.93 with standard deviation of 0.11.

**(Figure 1)**

The EBR results for the Skukuza eddy covariance system, with a mean of 0.93 (only the years with good data quality), are generally within the reported accuracies by most studies that report the energy balance closure error at 10 – 30%. Chen et al. (2009) report a mean of 0.98 EBR, average slope of 0.83, and $R^2$ ranges between 0.87 and 0.94 for their study in the semi-arid region of Mongolia. Wilson et al., (2002) also reported that the mean annual EBR for 22 FLUXNET sites was 0.84, ranging from 0.34 to 1.69, and slopes and intercepts ranging from 0.53 to 0.99, and from −33 to 37 W m$^{-2}$, respectively. Yuling et al. (2005) also report that in the ChinaFLUX,





EBR ranged between 0.58 and 1.00, with a mean of 0.83. von Randow et al. (2004) showed an energy
imbalance of 26 % even after correcting for the angle of attack on the sonic anemometer in the forested Jeru
study area in the Amazon, and explained this as due to either slow wind direction changes which result in low
frequency components that cannot be captured using short time rotation scales, and the difficulty in estimating
horizontal flux divergences caused by energy that is transported horizontally by circulations. Sanchez et al.,
(2010) showed that the inclusion of the storage term in the EBR improved the closure by almost 6 % from 0.72,
in their study in a FLUXNET boreal site in Finland.  Using data from the Tibetan Observation and Research
Platform (TORP), Liu et al. (2011) observed an EBR value of 0.85 in an alfalfa field in semi-arid China. Also
under similar semi-arid conditions, in China, an EBR value of 0.80 was found by Xin and Liu (2010) in a maize
crop. Were et al. (2007) reported EBR values of about 0.90 over shrub and herbaceous patches, in a dry valley
in southeast Spain.

### 3.1.2    Seasonal variation of EBR

Fig 2 shows the seasonal OLS results for the combined 15 year period. The slopes ranged between 0.99 and
1.28, with a mean of 1.17 ± 0.13, and the intercepts were a mean of 25.54 $Wm^{-2}$ ± 10.77 $Wm^{-2}$. $R^2$ ranged
between 0.73 and 0.82 with a mean of 0.78 ±0.05. The EBR for the different seasons ranged between 0.50 and
0.88, with a mean of 0.7. The winter season had the lowest EBR of 0.50, while the summer season had the
highest EBR of 0.88, autumn and spring had EBR of 0.68 and 0.74, respectively.
**(Figure 2)**
Wilson et al. (2002) comprehensively investigated the energy closure of the summer and winter seasons for 22
FLUXNET sites for 50 site-years. They also reported higher energy balance correlation during the summer
compared to the winter season, with the mean $R^2$ of 0.89 and 0.68, respectively. However, their EBR showed
smaller differences between the two seasons, being 0.81 and 0.72, for summer and winter, respectively, whereas
for Skukuza, the differences were much significant. Ma et al. (2009) reported an opposite result from the
Skukuza results, showing energy closures of 0.70 in summer and 0.92 in winter over the flat prairie on the
northern Tibetan Plateau.

### 3.1.3    Day – night-time effects

Fig 3 shows the daytime and nocturnal OLS regression results for the 15 year period. The daytime and nocturnal
slopes were 0.99 and 0.11, with the intercepts being 76.76 and 1.74 $Wm^{-2}$, respectively. Daytime and nocturnal
$R^2$ were 0.64 and 0.01, repectively. The EBR for the different times of day were 0.72 and -4.59, daytime and
nocturnal, respectively.

**(Figure 3)**

Results from other studies also reported a higher daytime surface energy balance closure. For instance, Wilson
et al., (2002) show that the mean annual daytime energy closure was 0.8, whereas the nocturnal EBR was
reported to be was negative or was much less or much greater than 1.





The large nocturnal energy imbalances are explained to be a result of low friction velocity, which leads to weak
turbulence. Lee and Hu (2002) hypothesized that the lack of energy balance closure during nocturnal periods
was often the result of mean vertical advection, whereas Aubinet et al., (1999) and Blanken et al., (1997)
showed that the energy imbalance during nocturnal periods is usually greatest when friction velocity is small.

## 3.2 Surface energy partitioning

### 3.2.1 Surface energy measurements

The daily mean measurements of the energy budget components from 2000 to 2014 are highlighted in Fig 4.
The seasonal cycle of each component can be seen throughout the years, where at the beginning of each year the
energy budget components are high, and as the each year progresses they all decrease to reach a low during the
middle of the year, which is the winter season. The multi-year daily means of Rn, H, LE and G were 139.1 $Wm^{-2}$,
57.70 $Wm^{-2}$, 42.81 $Wm^{-2}$ and 2.94 $Wm^{-2}$, with standard deviations of 239.75 $Wm^{-2}$, 104.15 $Wm^{-2}$, 70.58 $Wm^{-2}$
and 53.67 $Wm^{-2}$, respectively.
**(Figure 4)**

### 3.2.2 Influence of weather conditions and seasonality

In arid/semi-arid ecosystems, solar radiation is not a limiting factor for evapotranspiration, instead it is mainly
limited by water availability. The seasonal fluctuations of energy fluxes are affected by the seasonal changes in
the solar radiation, air and soil temperatures, and soil moisture (Baldocchi et al., 2000; Arain et al., 2003). These
climatic variables influence vegetation dynamics in an ecosystem, as well as how solar radiation is partitioned.
Hence, daily measurements of precipitation, soil moisture and air temperature for 2012 were evaluated to
investigate the partitioning of the surface energy in the semi-arid landscape of Skukuza.
Fig 5 presents daily averages of air temperature, soil water content and total precipitation for Skukuza
for 2012. The total annual precipitation was 534.24 mm, distributed from September and April, with the highest
monthly amount of 148.59 mm recorded in January. Soil water content ranged between 5.23 and 26.4 %, and
soil temperature varied between 18 and 30 °C. The mean daily air temperature shows some variability between
months, ranging between 9 and 32 °C, with the mean annual air temperature being 26 °C.
**(Figure 5)**
To illustrate the partitioning of solar radiation into the different fluxes throughout the year, Fig 6 presents the
multi-year mean monthly variations of the surface energy components. showing a general decrease of the
components between February and June, which then gradually increases again until November. The multi-year
monthly means of Rn, H, LE and G were 97.48 $Wm^{-2}$ (June) and 200.41 $Wm^{-2}$ (February), 34.59 $Wm^{-2}$ (June)
and 76.80 $Wm^{-2}$ (February), 7.06 $Wm^{-2}$ (July) and 104.02 $Wm^{-2}$ (January), -2.91 $Wm^{-2}$ and 21.55 $Wm^{-2}$
(September), respectively.
**(Figure 6)**
The higher monthly means of net radiation during the months of November and February as compared to those
of December and January is due to the presence of clouds and the fact that December and January are the





months of peak precipitation in the region (Scholes et al., 2001). Net radiation is affected by surface albedo,
presence of cloud and water vapour (Goosse et al., 2008).
Fig 7 illustrates the averaged diurnal variations of the surface energy balance components for the four
seasons (summer, autumn, winter and spring). The general trend reveals that sensible heat flux dominated the
energy partitioning during three seasons, followed by latent heat flux, and lastly the soil heat flux, except during
the summer season where latent heat flux was larger than sensible heat flux. This period is characterised by high
incoming solar radiation, as illustrated by the high midday net radiation of between 700 and 800 $Wm^{-2}$, and high
precipitation (Fig 5). Autumn (Fig 7b) is characterised by reduced net radiation, as shown by midday net
radiation of around 500 $Wm^{-2}$, whereas winter (Fig 7c) had the lowest midday net radiation, and minimum latent
heat flux.
**(Figure 7)**
Just before the first rains, i.e. between September and November, tree flowering and leaf emergence occurs in
the semi-arid savanna in the Skukuza area (Archibald and Scholes, 2007), and grasses shoot as soil moisture
availability improves with the rains (Scholes et al., 2003). This is characterised by a gradual increase in latent
heat flux (evapotranspiration), which, when compared to the winter season, is significantly lower than the
sensible heat flux, as illustrated in Fig 5 and 7. As the rainy season progresses, and vegetation development
peaks, latent heat flux also reaches its maximum, becoming significantly higher than sensible heat flux. Between
March and September, when leaf senescence occurs, the leaves gradually change colour to brown and grass to
straw, and trees defoliate, sensible heat flux again gradually becomes significantly higher than LE, as illustrated
in Fig 7b-d.
Gu et al. (2006) examined how soil moisture, vapour pressure deficit (VPD) and net radiation control
surface energy partitioning. They ascertained that with ample soil moisture, latent heat flux dominates over
sensible heat flux, and reduced soil moisture availability reversed the dominance of latent heat over sensible
heat, because of its direct effect on stomatal conductance. An increase in net radiation, on the other hand, also
increases both sensible and latent heat fluxes. The increase of either then becomes a function of soil moisture
availability, since they cannot increase in the same proportion. Gu et al., (2006) also revealed that the
relationship between net radiation and latent heat flux is convex, while that of net radiation and sensible heat
flux is concave. Their findings are consistent with our results, which show that during the rainy season, latent
heat flux was significantly higher than sensible heat flux, whereas, during the other seasons, sensible heat flux
remained higher than latent heat flux. The effect of vapour pressure deficit on energy partitioning is non-linear,
because of the opposing effects vapour pressure deficit exerts on latent heat flux. Li et al. (2012) also
investigated the partitioning of surface energy in the grazing lands of Mongolia, and concluded that the energy
partitioning was also controlled by vegetation dynamics and soil moisture availability, although soil heat flux is
reportedly higher than latent heat flux in most instances. In a temperate mountain grassland in Austria,
Harmmerle et al., (2008) found that the energy partitioning in this climatic region was dominated by latent heat
flux, followed by sensible heat flux and lastly soil heat flux.
The consensus in all above studies, including this one, is that vegetation dynamics play a critical role in
energy partitioning. They note that during full vegetation cover, latent heat flux is the dominant portion of net





radiation. However, depending on the climatic region, the limiting factors of energy partitioning vary between
water availability and radiation. Our study, thus, confirms that in semi-arid regions, sensible heat flux is the
highest fraction of net radiation throughout the year, except during the rainy summer period, when latent heat
flux surpasses sensible heat flux. However, in regions and locations where water availability is not a limiting
factor, latent heat flux may take the highest portion of net radiation.

### 3.2.3 Energy exchanges and inversions at night-day transitions
Fig 8 shows the turbulent fluxes normalised as fractions of net radiation. The diurnal variation of latent heat flux
at the site is characterized by a sharp cross-over from the negative to positive values around sunrise, and a return
to negative values after sunset and vice versa for soil heat flux in summer. The sharp rise in latent heat flux
experienced when the sun rises during summer is a result of the morning increases in net radiation and the
presence of dew, which evaporates as the sun heats the surface, and when the sun sets LE also drops to negative
values. Sensible heat flux remains constant throughout the day and night.  It is also evident that sensible heat
flux is dominant is all the seasons, except in summer. During wintertime, latent energy is negligible, and the
sensible heat flux is evidently more dominant.
**(Figure 8)**

## 4 Conclusion
This study investigated both surface energy balance and partitioning into latent, sensible and soil heat fluxes in a
semi-arid savanna ecosystem in Skukuza. 15 years of eddy covariance data analysis revealed the mean multi-
year energy balance ratio as 0.93, whereas the seasonal EBR varied between 0.50 and 0.88, with winter
recording the higher energy imbalance. Daytime EBR was as high as 0.72, with negative EBR for the nighttime.
The high energy imbalance at night was explained as a result of stable conditions, which limit turbulence that is
essential for the creation of eddies.
The energy partition analysis revealed that sensible heat flux is the dominant portion of net radiation in
this semi-arid region, except in summer, when precipitation falls. The results also show that water availability
and vegetation dymanics play a critical role in energy partitioning, whereby when it rains, vegetation growth
occurs, leading to an increase in  latent heat flux / evapotranspiration.
**Acknowledgements**
This study was supported by the Council for Scientific and Industrial Research under the project entitled
"Monitoring of water availability using geo-spatial data and earth observations", and the National Research
Foundation under the Thuthuka PhD cycle grant.

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





**Table 1: Measurements taken and instruments used at Skukuza flux tower**

| Instrument | Model/ brand | Measurement |
|---|---|---|
| Sonic anemometer | Gill Instruments Solent R3, Hampshire, England | 3-dimensional, orthogonal components of velocity (u, v, w ($ms^{-1}$)) |
| Closed path gas analyser | IRGA, LiCOR 6262, LiCOR, Lincoln | Water vapor, carbon dioxide concentrations |
| Radiometer | Kipp and Zonen CNR1, Delft, The Netherlands | Incoming and outgoing longwave and shortwave radiation |
| HFT3 plates | Campbell Scientific | Soil heat flux @ 5 cm depth |
| Frequency domain reflectometry probes | Campbell Scientific CS615, Logan, Utah | Volumetric soil moisture content @ different depths |




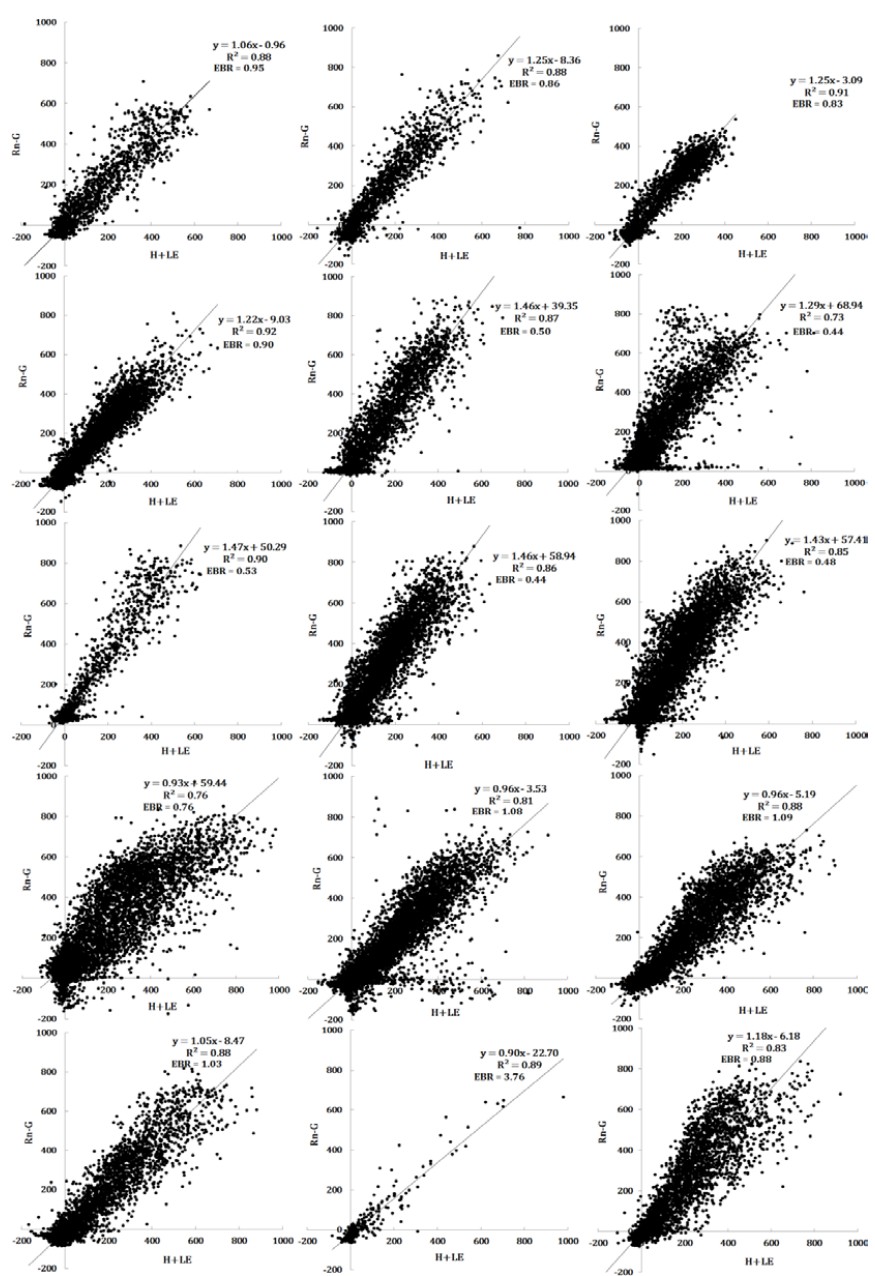

**Figure 1: 15-year series of annual regression analysis of turbulent (sensible and latent) heat fluxes against available energy (net radiation minus ground conduction heat) from 2000 to 2014 at Skukuza, (SA).**





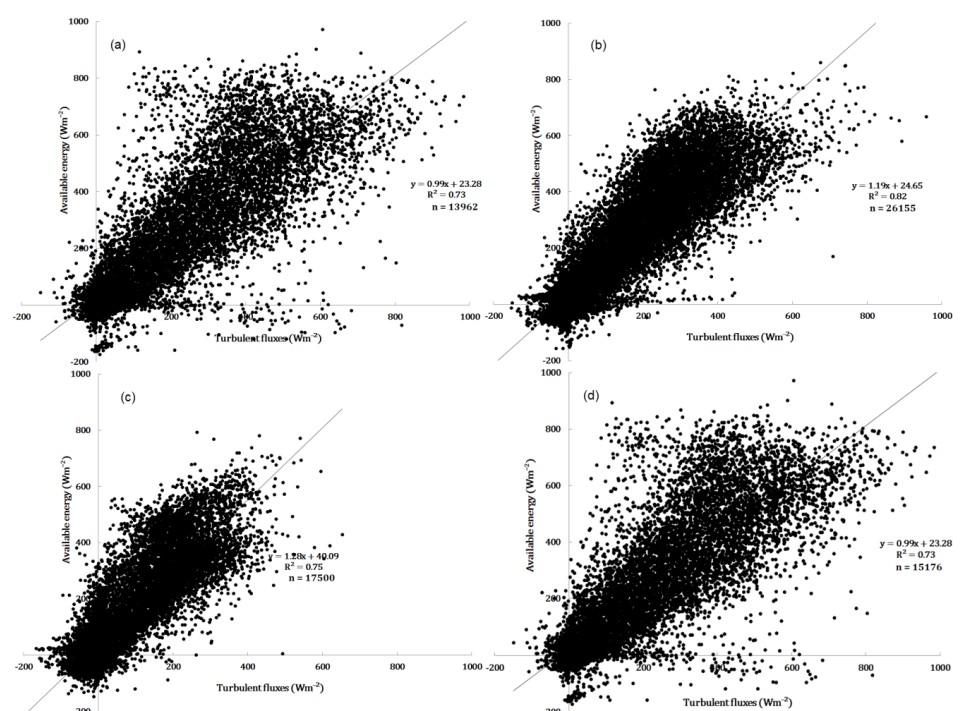

**Figure 2: Seasonal turbulent fluxes (H+LE) correlation to available energy (Rn-G) for Skukuza flux tower from (Dec-Feb (a), March-May (b), June-Aug (c), Sept-Nov (d))**




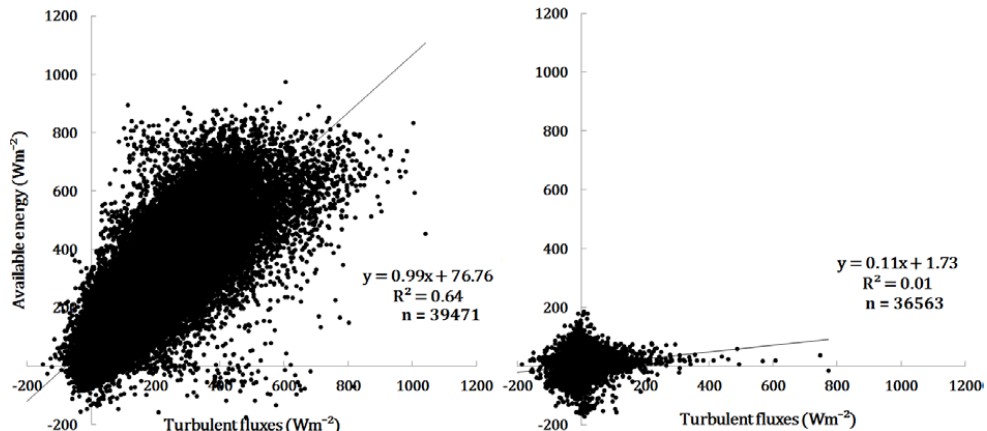


**Figure 3: Turbulent fluxes correlation to available energy for daytime (a) and night-time (b), using the full (2000-**
**2014) 15-year available data series**





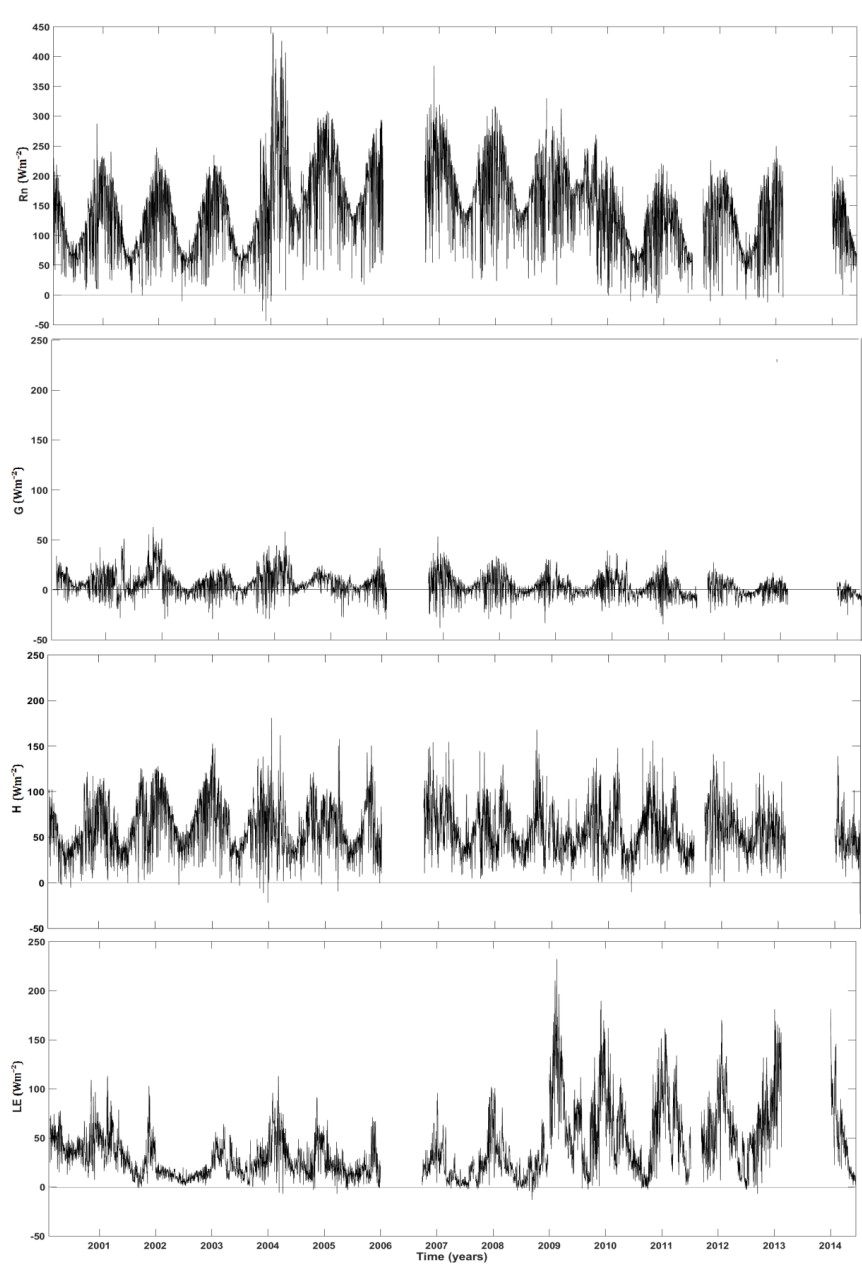


**Figure 4: Time series of daily mean surface energy balance component fluxes from 2000 to 2014 at Skukuza flux tower site (SA)**







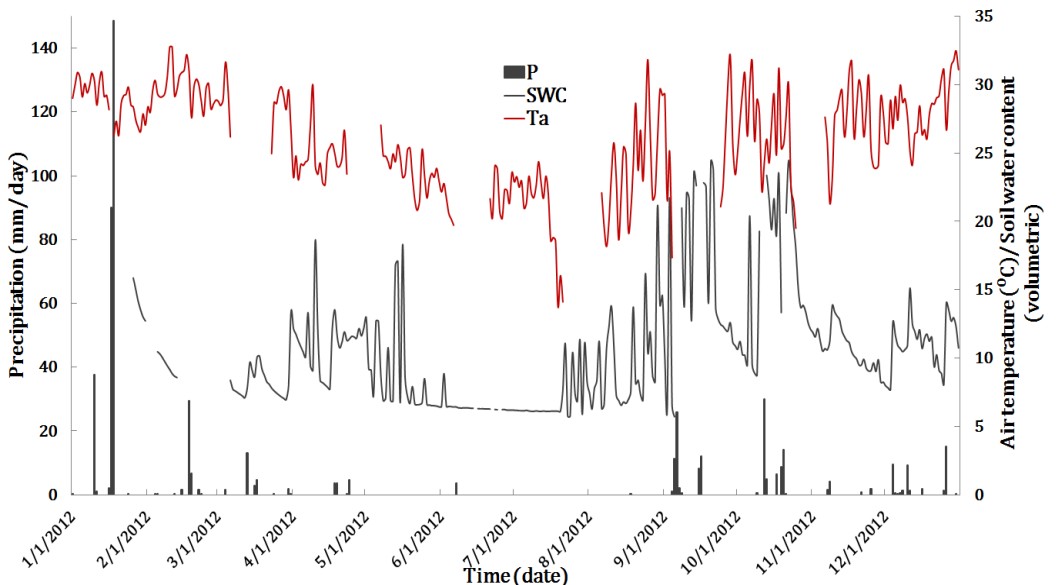


**Figure 5: Annual daily time series (2012) of meteorological measurements of mean air temperature, soil water content and precipitation from Skukuza flux tower station**






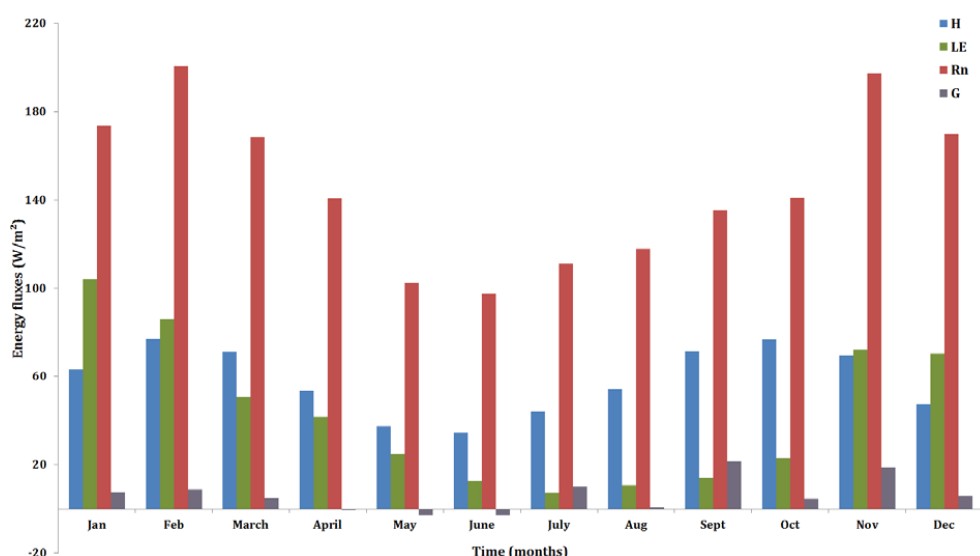


**Figure 6: 15-year (2000-2014) monthly means of surface energy balance fluxes of Skukuza flux tower site (SA)**




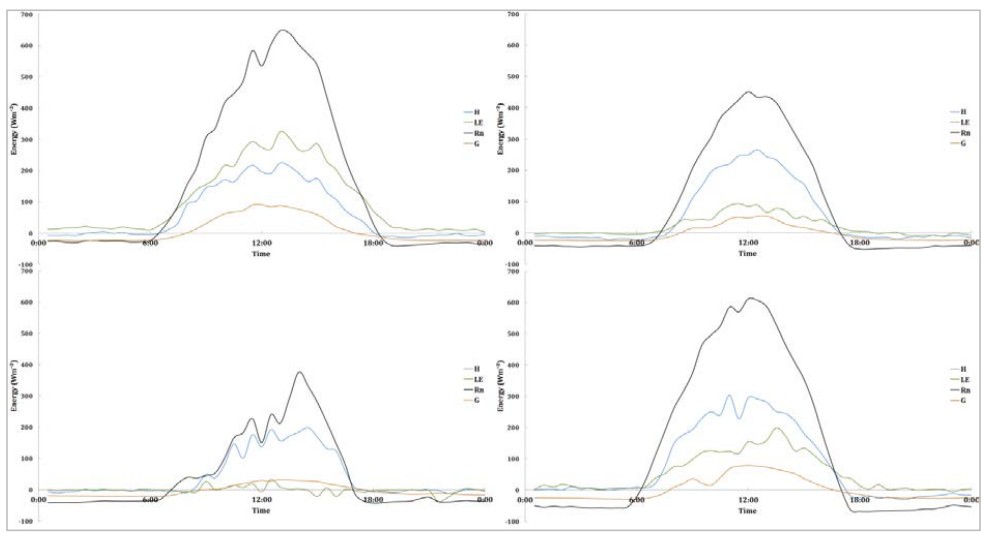

**Figure 7: Averaged diurnal surface energy balance component fluxes for the different seasons in 2012; summer (a), autumn (b), winter (c) and spring (d)**



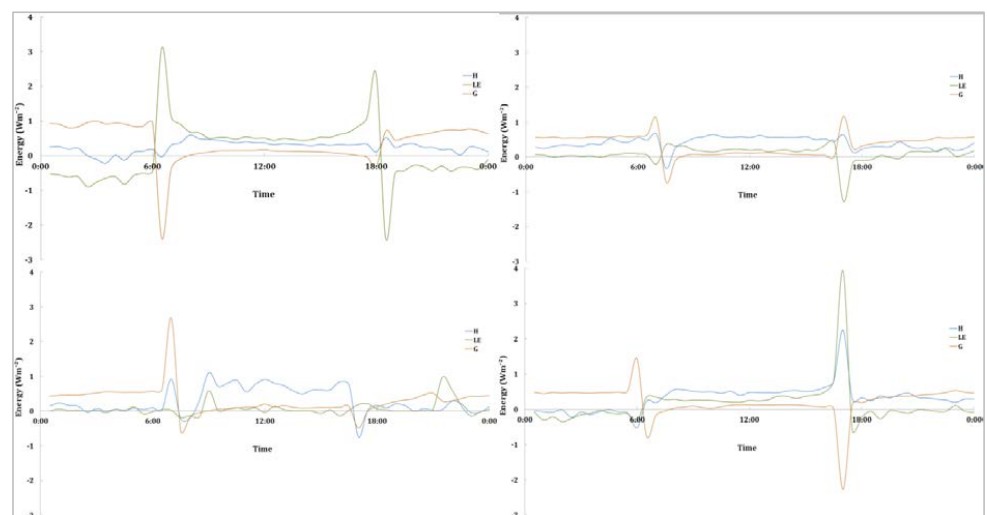

470

**Figure 8: Averaged normalised diurnal surface energy balance component variations for summer (top left), autumn (top left), winter (bottom left) and spring (bottom right) seasons**

473