# Peer review of "Analysing surface energy balance closure and partitioning 1 over a semi-arid savanna FLUXNET site in Skukuza, Kruger 2 National Park, South Africa 3"

_Hydrology and Earth System Sciences, 2016_

## Referee Comment (RC1) · C. Jimenez (Referee) · 21 Mar 2016

**General comments**

The paper presents a data record of 14 years of radiative and turbulent fluxes measured at a semi-arid savanna FLUXNET site in South Africa. The value of this data record cannot be discussed and it is great that a publication looks at energy partitioning for this type of environment (certainly not the environment where most tower fluxes are locate). Nevertheless, in my opinion the paper could have exploited the data record better and make more efforts to identify and separate possible instrumental issues from inter-annual variability. In general, the figures are not commented in great detail, and more efforts could have been made to make the paper more attractive to the readers. Some

of the papers mentioned with similar analyses (but in different environments, e.g., Gu et al., 2006 and Li et al., 2006) are more complete in that sense and propose more interesting analyses of the existing data.

Specific comments

P1-L16. The abstracts should be more specific about the paper findings, it seems more focused on just listing what the paper will be looking at.

P2-L67. The OLS could also be explained with a line if text or so a bit more in this context.

P2-L78. "Research on the South African savanna, i.e. using data from the Skukuza EC system", strange sentence, all research in South African savannah is linked to once EC system?

P2-L84. EBR is defined, but not EB on its own, presumably Energy Balance.

P4-L144. Not sure I understand the line "data without gaps". Does it refer to the original $\frac{1}{2}$ hour data being gap-filled, or to the seasonal averages?

P4-L154. I am confused here, what do random errors mean here? I have problems understanding that the Rn and G observations at the station are free from random errors, as I imagine that there is always some instrumental noise in the observations.

P4-L159. Potential to "remove"?

P5-L182. Mean of 1.19 +- 0.21, could you state what +-0.21 means?

P5-L183. Wm-? 2 missing?

P5-L184. The variation in the slopes and EBRs are scarily large. The authors are not looking for explanations? Assuming that the environmental conditions at the tower have not changed, and that the soil/vegetation covered by the fetch of the tower observations remains similar along the years, the variability has to be related to the effect

of missing data (not all years are sampled equally) and /or instrumental issues (e.g., instrumentation replacement). The latter is possibly more likely. For instance, I noticed that 2006-7-8 have slopes around 1.4, while 2009-10-11 around 0.9, with a change of Rn instruments in 2009. The authors should be looking into these things to help building confidence in the data record.

P5-L191. Absence of negative Rn-G because those times of the day were not measured, or because of issues with the instruments operating at those times? P6-L219. Figure 2 shows a larger number of outliers for summer and spring, any reasons for that?

P7-L242. The references point towards the EC measurements not being reliable at nighttime (low turbulence, advection, etc). What about the net radiation measurements at nighttime? More trustable than the EC ones?

P7-L252. There seems to be things to comment on Figure 4. What happens with the daily means in 2006? Why the Rn from 2004-2010 looks different form the other years? Inter-annual variability or instrumental issues? The LE, H, and G look more consistent from year to year.

P7-L275. Even if references are given, it will be good to explain the links between cloudiness and precipitation and the observed Rn seasonal variability. Clouds should increase the downward longwave component and reduce the downward shortwave. I'm not an expert, but it is not that obvious that the overall effect is an increase in the net radiation. Also, it may have helped to understand this figure to have Figure 5 plotted as monthly means, instead of a time series.

P8-L301. The findings of Gu 2006 correspond to a temperate forest site, so the environmental conditions are in principle different for the location of the study, which is a semi-arid savannah environment. It is worth mentioning.

P8-L302. The "concave" and "convex" mentioning requires further explanations, can it

be illustrated with the data at the Skukuza station? It does not seem obvious from the figures shown so far in the paper, or I am missing something.

P9-L321. I am having some problems understanding Figure 8. If the aim is to discuss the partitioning of the heat fluxes, perhaps it could have been better to normalize with the available energy, i.e., the ratio of LE and H with Ae=Rn-G and only plot LE/Ae and H/Ae. This is because the energy closure shown in the figure seems very poor some times, so I am wondering if we can draw any conclusions about the fluxes partitioning at those times of the day. If we are plotting LE/Rn, H/Rn, and G/Rn, the sum LE/Rn+H/Rn+G/Rn = Rn/Rn =1 if energy closure was perfect and there were no missing terms. Now, if we take just before 18 hours in spring, LE/Rn∼=4, H/Rn∼=2, G/Rn∼=-2, so the net sum is 4 instead of 1 (for perfect closure). Or, in other words, the energy required for that situation is 4 times larger than the available Rn. A similar thing happens in summer around the same time, in winter around 6 hours. Is there a source of energy missing, or is it related to instrumental issues (small value of the fluxes and ratios between them)?

P9-L325. In summer before the sun sets, there is a new peak of positive LE not too different in magnitude from the peak associated to the presence of dew. What can be the cause for that?

P9-L332. The conclusions are too short and too general. A food example is the last sentence "The results also show that water availability
and vegetation dymanics play a critical role in energy partitioning, whereby when it rains, vegetation growth
occurs, leading to an increase in latent heat flux / evapotranspiration", which is certainly true, but sort of common knowledge.

Table 1. Any specific reasons to replace "at" by "@" in the text of the Table? Figure 1. Years should be added to the individual plots.

Figure 2. For consistency with Figure 1, it would be more useful to have the EBR in the plots, instead of the number of points.

Figure 5. Is air humidity also measured at the station in a routine basis? Given the study of the heat flux partitioning, something like VPD would have been nice to have and analyse. Figure 6. It would have been nice to have a new bar with the H+LE+G, so it could be compared with Rn and used to assess the seasonal energy balance closure.

Figure 7. The labels, legends, and lines are difficult to read, they need to be made larger. The a, b, c, d symbols are missing in the figures.

Figure 8. Same as figure 7, we can hardly read the labels or identify the colours of the lines.

---

## Referee Comment (RC2) · Anonymous Referee #2 · 29 Mar 2016

General comments The authors evaluated a 15-year EC data record of a savanna FLUXNET site. The authors focus in their analysis on the surface energy balance closure and energy partitioning. Among others they derive monthly mean surface energy fluxes and energy balance ratios (EBR) over the last 15-years. The topic fits very well into the scope of HESS, and the dataset would be interesting for a broad readership of HESS. My main criticism is that the data analysis remains in large parts superficial and does not go into depth. That is also the reason why at the end the authors do not come to really novel conclusions. There is no question that this is a great and unique dataset. This dataset must be published, but this dataset also deserves an in-depth analysis. Many questions remain open: What was the reason for the huge systematic

errors in the flux measurements of the years 2004-2008 and 2013? Do the data show any long-term trend with regard to the energy and water fluxes? Are the EC flux data sufficiently accurate to be used to validate satellite-based evapotranspiration methods? What's about the uncertainty range of the flux data due to the lack of the closure of the energy balance? Which years were the extreme years with regard to temperature and rainfall? Etc. Before publication the authors must spend significantly more efforts in the analysis of this great dataset!

Specific comments Line 21-24: This paragraph is not really needed in the Abstract.

Line 27: Please cancel the word concept.

Line 39: Because there are besides canopy and ground heat storage other minor flux terms, I suggest to rewrite the sentence as follows: ". . . (G) heat fluxes and other minor flux terms such as heat stored by the canopy and ground."

Line 65: Please rewrite as follows: ". . . is an accepted performance criterion of EC flux data . . .". Remark: Please use everywhere the introduced abbreviations.

Line 86: The abbreviation EB was not introduced before.

Table 1: Please state here the number of replicates of the soil heat flux measurements. And it remains unclear, whether the authors considered the soil heat storage change in the upper 5-cm layer in their calculation of the soil heat flux at the surface. Please explain!

Table 1: Please state the exact installation depths of the CS615 probes.

Line 117: Please explain here or in the Discussion why you decided to switch from a closed-path to an open-path gas analyzer. Is the change in the instrumentation somehow related to the performance of the EC system?

Line 120: As in line 117: Please explain why you did not continue to measure net radiation with the CNR2 sensor.

Line 157: The summation sign is not needed here. Write the equation simply as (H+LE)/(Rn-G).

Line 174: The authors should give at the beginning of the Results section an overview of the weather conditions over the last 15 years. Which years were particularly dry or wet. Which years were particularly warm or cold. Did you observe any long-term trend in the weather data.

Line 176: From Figure 1 I would expect that the year 2013 was the year with the largest number of missing values and not the year 2001. Please explain.

Line 214: What do you mean here with combined? Please explain. Did you not exclude here the years with low EBRs? Why does the mean EBR here does not agree with the figure (0.93) you gave in chapter 3.1.1?

Line 236-242: Here it is not sufficient to explain the low EBR over the night time by referring to other studies. Please check you statement/conclusion against your own data. Are low EBRs related to low friction velocities?

Line 249: I think it would be better to compile all the numbers in a table, and please do not aggregate the data to multi-year daily means. You lose so much information. The authors should think about, for example, to give mean, minimum, and maximum monthly fluxes for every year.

Line 255-260: Why did you limit this analysis to the year 2012? Please explain.

Line 270-272: Here the meaning of the months in brackets remains unclear. Also here it would be better to compile the data in a table. Please give in this context also the Bowen ratios.

Figure 6: I suggest to plot the data as stacked columns and to include in the figure the residual! In this context it would be also important to give the possible range of fluxes due this residual (see e.g. Falge et al., 2005; Ingwersen et al., 2015) and to discuss whether the residual hampers the use of the data to validate satellite-based

evapotranspiration methods.

Line 338: This sentence reads strange. The sensible heat flux is not a part of net radiation. Please rewrite.

Line 331: Please revise the conclusions. They remain too general and on the level of text book knowledge. There must be something novel that we can learn from this 15-year long-term EC data record.

References: Falge, E., Reth, S., Brüggemann, N., Butterbach-Bahl, K., Goldberg, V., Oltchev, A., Schaaf, S., Spindler, G., Stiller, B., Queck, R., Köstner, B., and Bernhofer, C. (2005): Comparison of surface energy exchange models with eddy flux data in forest and grassland ecosystems of Germany, Ecol. Model., 188, 174–216.

Ingwersen, J., Imukova, K., Högy, P., Streck, T. (2015): On the use of the post-closure methods uncertainty band to evaluate the performance of land surface models against eddy covariance flux data, Biogeosciences, 12 (8), pp. 2311-2326.

---

## Referee Comment (RC3) · N. van de Giesen (Referee) · 11 Apr 2016

Review hess-2016-76

General comments

The main strength of the paper is, as the other two reviewers remarked, the long term dataset from the African savanna. For me, this is in itself sufficient although I would also prefer to see a better analysis. It may be difficult, however, to say much more than what has been said without becoming speculative. So if a deeper analysis is possible, I would definitely recommend that. If that can only be done in a speculative way, then the value would not be large. What should happen is publication of the dataset itself so

everyone can use it from now on. That would make the value so much higher.

Besides this, my main concern is that the ground heat flux G is, as usual, the step child of the energy balance. I understand that over longer periods, G becomes negligible but nothing is said about how it is measured at all. G may be part of the night time problems. Please describe the method. Heatflux plates are mentioned in a table but need to be in text as well. Heatflux plates are not really a good way to measure soil heatflux over any area (see, for example gentine et al in doi 10.1029/2010WR010203 and Jansen et al in GRACE, Remote Sensing and Ground-based Methods in Multi-Scale Hydrology (Proceedings of Symposium J-H01 held during IUGG2011 in Melbourne, Australia, July 2011) (IAHS Publ. 343, 2011)). Over periods of 30', G can be very important.

Minor comments

It is a bit a matter of taste but the word 'evapotranspiration' is not a happy one. See doi 10.1002/hyp.5563 for arguments.

21: 'extent'

26: Not sure what is meant with 'under development' here. Seems vague and does not add information.

29: Introduce 'EB' at first use of energy balance.

38: Leave out: 'for transformation [. . .] i.e.'

49: Leave out 'Hence'

57: Change to: 'the measured available energy'

62: Is high frequency transport also not underestimated?

80: Replace 'Hence, the need to' with 'Here, we' (the 'hence' was not really a logical connection,

82: 15 years: This is really a unique aspect and should also enter the abstract etc.

151: 'evaluated at different'

177: The standard deviation is not really something of interest here, I would think.

187: The range is not described well as 2013 is not part of it.

223: Summer & winter is a bit confusing here. Later it becomes clear which months are which but as summer is hot&wet and winter is warm&dry, it differs from what many other places experience as summer & winter. Perhaps better stick to wet & dry season.

248: 'and as each'

261: This paragraph and associated figure is not helpful. There is no comparison between weather and results (may be the most obvious point of entry to deepen the analysis) so just a climate picture does not help the reader. As mentioned before, the data should be made available on-line.

315: Here and elsewhere, it is not clear why the examples from the literature were chosen. One could expect more examples from the savanna or a structural overview of different climates but now it seems a bit random.

321: Please rethink this part. I agree that the transitions are indeed interesting, it becomes difficult to interpret with this normalization. It is said that 'sensible heat flux is dominant' etc but when the net radiation is near zero, the normalization does strange things and that is all the figures then say.

Figure 1,2,3: please use 'heat plots", the ones where you see small individual points where there is space and where the color changes from blue to red depending on the density of the dots where they can no longer be discerned.

Figures 7 8: Bigger lettering

---

## Short Comment (SC1) · 26 May 2016

**Response to reviewer 3 comments**

Thank you for taking time to review our manuscript.

We thank Reviewer#3 for reviewing of this manuscript, and for contributing to its improvement. We tried to answer every specific comment in detail as shown below:

**Minor comments**

*It is a bit a matter of taste but the word 'evapotranspiration' is not a happy one. See doi10.1002/hyp.5563 for arguments.*

Agreed.

21: 'extent'

26: Not sure what is meant with 'under development' here. Seems vague and does not add information.

The abstract has been rewritten as:

L16-35: Flux towers provide essential terrestrial climate, water and radiation budget information needed for environmental monitoring and evaluation of climate change impacts on ecosystems and society in general. They are also intended for calibration and validation of satellite-based earth observation and monitoring efforts, such as assessment of evapotranspiration from land and vegetation surfaces using surface energy balance approaches.

In this paper, 15 years of Skukuza eddy covariance data, i.e. from 2000 to 2014, were analysed for surface energy balance closure and partitioning. The surface energy balance closure was evaluated using the ordinary least squares regression (OLS) of turbulent energy fluxes (sensible (H) and latent heat (LE)) against available energy (net radiation (Rn) less soil heat (G)). Partitioning of the surface energy during the wet and dry seasons was investigated, as well as how it is affected by atmospheric vapor pressure deficit (VPD), and net radiation.

After filtering years with bad data, our results show an overall mean surface energy balance closure of 0.93. Seasonal variations of EBR also showed summer had best EBR with winter having the least closure. Nocturnal surface energy closure was lowest, and this was linked to low friction velocity during night-time, and an increase in friction velocity showed an increase in closure. The high surface energy balance closure gives confidence on the usability of these data for calibrating and validating

The surface energy partitioning of this savanna ecosystem showed that sensible heat flux dominated the energy partitioning between March and October, followed by latent heat flux, and lastly the soil heat flux, except during the wet season where latent heat flux was larger than sensible heat flux. An increase in net radiation was characterised by an increase in both LE and H, with LE showing a higher rate of increase than H in the wet season, and the reverse happening during the dry season. An increase in VPD is characterised by a decrease in LE and increase in H during the wet season, and an increase of both fluxes during the dry season.

29: Introduce 'EB' at first use of energy balance.

Introduced.

*38: Leave out: 'for transformation [: : :] i.e.'*

Done.

*49: Leave out 'Hence'*

Done.

*57: Change to: 'the measured available energy'*
Done.

*62: Is high frequency transport also not underestimated?*
Thank you for the observation.

*80: Replace 'Hence, the need to' with 'Here, we' (the 'hence' was not really a logical connection*
Thank for the comment.
The sentence has been rewritten as:
L79-80: However, there has been no investigation of surface energy partitioning and energy balance closure in this ecosystem.

*82: 15 years: This is really a unique aspect and should also enter the abstract etc.*
Thank you for the comment. Noted.

*151: 'evaluated at different'*
Corrected.

*177: The standard deviation is not really something of interest here, I would think.*
Noted, thank you.

*187: The range is not described well as 2013 is not part of it.*
Thank you for the comment.
The opening sentence (L202-203) states the range as between 0.44 in 2007 and 3.76 in 2013.

*223: Summer & winter is a bit confusing here. Later it becomes clear which months are which but as summer is hot&wet and winter is warm&dry, it differs from what many other places experience as summer & winter. Perhaps better stick to wet & dry season.*
Thank you for the comment. Noted.

*248: 'and as each'*
Noted, thank you.

*261: This paragraph and associated figure is not helpful. There is no comparison between weather and results (may be the most obvious point of entry to deepen the analysis) so just a climate picture does not help the reader. As mentioned before, the data should be made available on-line.*
Thank you for the comment. This subsection has been removed and further analysis done.
L321-333: The influence of VPD and Rn on surface energy partitioning was investigated during the wet and dry seasons. Results show that there is an increase in H and decrease in LE with an increase in VPD in the wet season (Fig 9). As illustrated earlier (Fig 1), VPD is higher when there is little or no rain (low soil water availability), which explains the increase in H with a rise VPD. In this instance, although the evaporative demand is high, the stomatal conductance is reduced due to absence of water in the soil, resulting in smaller LE and higher H. Rn, on the other hand, is partitioned into different fluxes, based on other climatic and vegetation physiological characteristics. Figure 10 illustrates that both latent and sensible heat flux increase with increase in net radiation, although their increases are not in proportion. During the wet season, the rate of increase of LE is higher than that of H, whereas in the dry season the reverse is true. The rate of increase of LE is controlled by the availability of soil water (precipitation), and during the wet season it increases steadily with increasing Rn, resulting in a convex, whereas the rate of

increase of H is concave, showing saturation with an increase in Rn. The opposite is true during the dry season, with limited water availability, the rate of increase of LE slows down with increase in Rn giving a concave, and a steady increase of H with Rn increase.

*315: Here and elsewhere, it is not clear why the examples from the literature were chosen. One could expect more examples from the savanna or a structural overview of different climates but now it seems a bit random.*

I have included an analysis of surface energy partitioning similar to Gu et al. (2006).

L320-332: The influence of VPD and Rn on surface energy partitioning was investigated during the wet and dry seasons. Results show that there is an increase in H and decrease in LE with an increase in VPD in the wet season (Fig 9). As illustrated earlier (Fig 1), VPD is higher when there is little or no rain (low soil water availability), which explains the increase in H with a rise VPD. In this instance, although the evaporative demand is high, the stomatal conductance is reduced due to absence of water in the soil, resulting in smaller LE and higher H. Rn, on the other hand, is partitioned into different fluxes, based on other climatic and vegetation physiological characteristics. Figure 10 illustrates that both latent and sensible heat flux increase with increase in net radiation, although their increases are not in proportion. During the wet season, the rate of increase of LE is higher than that of H, whereas in the dry season the reverse is true. The rate of increase of LE is controlled by the availability of soil water (precipitation), and during the wet season it increases steadily with increasing Rn, resulting in a convex, whereas the rate of increase of H is concave, showing saturation with an increase in Rn. The opposite is true during the dry season, with limited water availability, the rate of increase of LE slows down with increase in Rn giving a concave, and a steady increase of H with Rn increase.

*321: Please rethink this part. I agree that the transitions are indeed interesting, it becomes difficult to interpret with this normalization. It is said that 'sensible heat flux is dominant' etc but when the net radiation is near zero, the normalization does strange things and that is all the figures then say.*

This section has been removed.

*Figure 1,2,3: please use 'heat plots", the ones where you see small individual points where there is space and where the color changes from blue to red depending on the density of the dots where they can no longer be discerned.*

Thank you for the comment.

The authors do not think there is any information lost by using the normal figures as they have used.

*Figures 7 8: Bigger lettering*

They have been removed.

---

## Author Comment (AC1) · 26 May 2016

We thank Reviewer#1 for the positive evaluation of this manuscript, and for having contributed to its improvement. According to his general comments, extra evaluations were done to ascertain whether the variation in EBR results that we found were instrument-related or due to seasonality and weather conditions. Further analysis was also done to assess surface energy balance partitioning as influenced net radiation and vapour pressure deficit. We hope that this effort will improve the manuscript, by strengthening the weak points highlighted by the Reviewer. We tried to answer every specific comment in detail.

[Figure]

Please also note the supplement to this comment:
http://www.hydrol-earth-syst-sci-discuss.net/hess-2016-76/hess-2016-76-AC1-supplement.pdf

[Figure]

Plot a:
y = 0.94x - 7.97
$R^2$ = 0.84
EBR = 1.12
n = 10275

Plot b:
y = 1.13x + 6.16
$R^2$ = 0.87
EBR = 0.84
n = 20129

Plot c:
y = 1.21x + 14
$R^2$ = 0.81
EBR = 0.70
n = 10012

Plot d:
y = 1.10x - 14
$R^2$ = 0.88
EBR = 1.02
n = 8232

**Fig. 1.** Figure 3: Seasonal turbulent fluxes correlation to available energy for Skukuza flux tower from summer, (a), autumn (b), winter (c), spring (d)

**Supplement:**

**Response to Reviewer 1 comments**

We thank Reviewer#1 for the positive evaluation of this manuscript, and for having contributed to its improvement. According to his general comments, extra evaluations were done to ascertain whether the variation in EBR results that we found were instrument-related or due to seasonality and weather conditions. Further analysis was also done to assess surface energy balance partitioning as influenced net radiation and vapour pressure deficit. We hope that this effort will improve the manuscript, by strengthening the weak points highlighted by the Reviewer. We tried to answer every specific comment in detail as shown below:

*P1-L16. The abstracts should be more specific about the paper findings, it seems more focused on just listing what the paper will be looking at.*
The abstract now reads:
L16-35: Flux towers provide essential terrestrial climate, water and radiation budget information needed for environmental monitoring and evaluation of climate change impacts on ecosystems and society in general. They are also intended for calibration and validation of satellite-based earth observation and monitoring efforts, such as assessment of evapotranspiration from land and vegetation surfaces using surface energy balance approaches.

In this paper, 15 years of Skukuza eddy covariance data, i.e. from 2000 to 2014, were analysed for surface energy balance closure and partitioning. The surface energy balance closure was evaluated using the ordinary least squares regression (OLS) of turbulent energy fluxes (sensible (H) and latent heat (LE)) against available energy (net radiation (Rn) less soil heat (G)). Partitioning of the surface energy during the wet and dry seasons was investigated, as well as how it is affected by atmospheric vapor pressure deficit (VPD), and net radiation.

After filtering years with bad data, our results show an overall mean surface energy balance closure of 0.93. Seasonal variations of EBR also showed summer had best EBR with winter having the least closure. Nocturnal surface energy closure was lowest, and this was linked to low friction velocity during night-time, and an increase in friction velocity showed an increase in closure. The high surface energy balance closure gives confidence on the usability of these data for calibrating and validating

The surface energy partitioning of this savanna ecosystem showed that sensible heat flux dominated the energy partitioning between March and October, followed by latent heat flux, and lastly the soil heat flux, except during the wet season where latent heat flux was larger than sensible heat flux. An increase in net radiation was characterised by an increase in both LE and H, with LE showing a higher rate of increase than H in the wet season, and the reverse happening during the dry season. An increase in VPD is characterised by a decrease in LE and increase in H during the wet season, and an increase of both fluxes during the dry season.

*P2-L67. The OLS could also be explained with a line if text or so a bit more in this context*

The sentence now reads:

P2-L64-67: The surface energy balance closure is an accepted validation procedure of eddy covariance data quality (Twine et al., 2000; Wilson et al., 2002), and different methods have been used to assess the energy closure and partitioning, including ordinary least squares regression (OLS) method, i.e. a plot of turbulence fluxes (H+LE) against available energy (Rn-G), the residual method, i.e. Residual = Rn-G-H-LE, and the energy balance ratio, i.e. EBR = LE+H/Rn-G.

*P2-L78. "Research on the South African savanna, i.e. using data from the Skukuza EC system", strange sentence, all research in South African savannah is linked to once EC system?*

This sentence has been revised:

P2-L76-77: Research using data from the Skukuza EC system has focused mainly on the carbon exchange, fire regimes, and in global analysis of the energy balance…

*P2-L84. EBR is defined, but not EB on its own, presumably Energy Balance.*

P2-L82: energy balance added.

*P4-L144. Not sure I understand the line "data without gaps". Does it refer to the original ½ hour data being gap-filled, or to the seasonal averages?*

Deleted the sentence.

*P4-L154. I am confused here, what do random errors mean here? I have problems understanding that the Rn and G observations at the station are free from random errors, as I imagine that there is always some instrumental noise in the observations.*

The sentence has been modified, and now reads:

P4-L152-154: This method is only valid when there are no random errors in the independent variables, i.e. Rn and G, which of course is an incorrect assumption.

*P4-L159. Potential to "remove"?*

P4-L158: Neglect has been replaced by remove as recommended.

*P5-L182. Mean of 1.19 +- 0.21, could you state what +-0.21 means?*

± means standard deviation.

*P5-L183. Wm-? 2 missing?*

Corrected.

*P5-L184. The variation in the slopes and EBRs are scarily large. The authors are not looking for explanations? Assuming that the environmental conditions at the tower have not changed, and that the soil/vegetation covered by the fetch of the tower observations remains similar along the years, the variability has to be related to the effect of missing data (not all years are sampled equally) and /or instrumental issues (e.g., instrumentation replacement). The latter is possibly more likely. For instance, I noticed that 2006-7-8 have slopes around 1.4, while 2009-10-11 around 0.9, with a change of Rn instruments in 2009. The authors should be looking into these things to help building confidence in the data record.*

Thank you for the comment. The explanation has been given as:

L205-211: Between 2000 and 2004, the CNR2 net radiometer was used to measure long and shortwave radiation, and these were combined to derive Rn. However, when the pyrgeometer broke down in 2004, Rn was derived from measured shortwave radiation and modelled longwave radiation until the CNR2 was replaced by the NRLite net radiometer in 2009. This was a high source of error, as shown by the low EBR between 2004 and 2008. The closed-path gas analyser was also changed to open-path gas analyser in 2006. An analysis of the 2006 data (which had very low data completeness of 7.59 %) showed that there were no measurements recorded until September, possibly due to instrument failure.

*P5-L191. Absence of negative Rn-G because those times of the day were not measured, or because of issues with the instruments operating at those times?*

Thank you for the comment.

Please see the above response.

*P6-L219. Figure 2 shows a larger number of outliers for summer and spring, any reasons for that?*

Thank you for the comment.

For the seasonal EBR assessment, we had not filtered out the 2004-2008 data; after doing so, we saw improved results (see Fig 3).

L238-241: A large number of outliers is observed in summer due to weather conditions like clouds and rainfall events that make the thermopile surface wet, thus reducing the accuracy of the net radiometer. A study comparing different the performance of different net radiometers by Blonquist et al. (2009) shows that the NR-Lite is highly sensitive to precipitation and dew/ frost since it the sensor is not protected.

**(Figure 1)**

*P7-L242. The references point towards the EC measurements not being reliable at night-time (low turbulence, advection, etc). What about the net radiation measurements at night-time? More trustable than the EC ones?*

Thank you for your comment.

L271-275: Another source of error in the nocturnal EBR is the high uncertainty in night-time measurements of Rn. At night, the assumption is that there is no shortwave radiation, and Rn is a product of longwave radiation. Studies show that night-time measurements of longwave radiation were less accurate than daytime measurements (Blonquist et al., 2009). The RN-Lite, for instance has low sensitivity to longwave radiation, resulting in low accuracy in low measurements.

*P7-L252. There seems to be things to comment on Figure 4. What happens with the daily means in 2006? Why the Rn from 2004-2010 looks different form the other years? Inter-annual variability or instrumental issues? The LE, H, and G look more consistent from year to year.*

Thank you for the comment.

L284-287: The gap in 2006 indicates the absence of the surface energy measurements in that year, a result of instrument failure. Between 2004 and 2008, the Rn was calculated as a product of measured shortwave radiation and modelled longwave radiation, which was a high source of error in the estimation of Rn. These years are also characterised by low EBR.

*P7-L275. Even if references are given, it will be good to explain the links between cloudiness and precipitation and the observed Rn seasonal variability. Clouds should increase the downward longwave component and reduce the downward shortwave. I'm not an expert, but it is not that obvious that the overall effect is an increase in the net radiation. Also, it may have helped to understand this figure to have Figure 5 plotted as monthly means, instead of a time series.*

Thank you for the question.

We removed the section.

*P8-L301. The findings of Gu 2006 correspond to a temperate forest site, so the environmental conditions are in principle different for the location of the study, which is a semi-arid savannah environment. It is worth mentioning.*

Thank you. The sentence now reads:

L333-334: Gu et al. (2006) examined how soil moisture, vapour pressure deficit (VPD) and net radiation control surface energy partitioning at a temperate deciduous forest site in central Missouri, USA.

*P8-L302. The "concave" and "convex" mentioning requires further explanations, can it be illustrated with the data at the Skukuza station? It does not seem obvious from the figures shown so far in the paper, or I am missing something.*

Thank you for the comment.

L320-332: The influence of VPD and Rn on surface energy partitioning was investigated. Results show that there is an increase in H and decrease in LE with an increase in VPD in the wet season (Fig 9).As illustrated earlier (Fig 1), VPD is higher when there is little or no rain (low soil water availability), which explains the increase in Bowen ratio with a rise VPD. In this instance, although the evaporative demand is high, the stomatal conductance is reduced due to absence of water in the soil, resulting in smaller LE and higher H, and thus higher Bowen ratio. Rn, on the other hand, is partitioned into different fluxes, based on other climatic and vegetation physiological characteristics. Figure 10 illustrates that both latent and sensible heat flux increase with increase in net radiation, although their increases are not in proportion. During the wet season, the rate of increase of LE appears to be higher than that of H, whereas in the dry season the reverse is true. The rate of increase of LE is controlled by the availability of soil water (precipitation), and during the wet season it increases steadily with increasing Rn, resulting in a convex, whereas the rate of increase of H is concave, showing saturation with an increase in Rn. The opposite is true during the dry season, with limited water availability, the rate of increase of LE slows down with increase in Rn giving a concave, and a steady increase of H with Rn increase.

*P9-L321. I am having some problems understanding Figure 8. If the aim is to discuss the partitioning of the heat fluxes, perhaps it could have been better to normalize with the available energy, i.e., the ratio of LE and H with Ae=Rn-G and only plot LE/Ae and H/Ae. This is because the energy closure shown in the figure seems very poor sometimes, so I am wondering if we can draw any conclusions about the fluxes partitioning at those times of the day. If we are plotting LE/Rn, H/Rn, and G/Rn, the sum LE/Rn+H/Rn+G/Rn = Rn/Rn =1 if energy closure was perfect and there were no missing terms. Now, if we take just before 18 hours in spring, LE/Rn_=4, H/Rn_=2, G/Rn_=-2, so the net sum is 4 instead of 1 (for perfect closure). Or, in other words, the energy required for that situation is 4 times larger than the available Rn. A similar thing happens in summer around the same time, in winter around 6 hours. Is there a source of energy missing, or is it related to instrumental issues (small value of the fluxes and ratios between them)?*

This section was removed.

*P9-L325. In summer before the sun sets, there is a new peak of positive LE not too different in magnitude from the peak associated to the presence of dew. What can be the cause for that?*

*P9-L332. The conclusions are too short and too general. A food example is the last sentence "The results also show that water availability land vegetation dynamics play a critical role in energy partitioning, whereby when it rains, vegetation growth, leading to an increase in latent heat flux / evapotranspiration", which is certainly true, but sort of common knowledge.*

*Table 1. Any specific reasons to replace "at" by "@" in the text of the Table? Figure 1. Years should be added to the individual plots.*
Corrected.

*Figure 2. For consistency with Figure 1, it would be more useful to have the EBR in the plots, instead of the number of points.*
Done, see Fig 3.

*Figure 5. Is air humidity also measured at the station in a routine basis? Given the study of the heat flux partitioning, something like VPD would have been nice to have and analyse. Figure 6. It would have been nice to have a new bar with the H+LE+G, so it could be compared with Rn and used to assess the seasonal energy balance closure.*
Thank you for your comment. We have analysed how VPD influences surface energy partitioning:
L320-332: The influence of VPD and Rn on surface energy partitioning was investigated during the wet and dry seasons. Results show that there is an increase in Bowen ratio with an increase in VPD in the wet season (Fig 9). As illustrated earlier (Fig 1), VPD is higher when there is little or no rain (low soil water availability), which explains the increase in Bowen ratio with a rise VPD. In this instance, although the evaporative demand is high, the stomatal conductance is reduced due to absence of water in the soil, resulting in smaller LE and higher H, and thus higher Bowen ratio. Rn, on the other hand, is partitioned into different fluxes, based on other climatic and vegetation physiological characteristics. Figure 10 illustrates that both latent and sensible heat flux increase with increase in net radiation, although their increases are not in proportion. During the wet season, the rate of increase of LE appears to be higher than that of H, whereas in the dry season the reverse is true. The rate of increase of LE is controlled by the availability of soil water (precipitation), and during the wet season it increases steadily with increasing Rn, resulting in a convex, whereas the rate of increase of H is concave, showing saturation with an increase in Rn. The opposite is true during the dry season, with limited water availability, the rate of increase of LE slows down with increase in Rn giving a concave, and a steady increase of H with Rn increase.

*Figure 7. The labels, legends, and lines are difficult to read, they need to be made larger. The a, b, c, d symbols are missing in the figures.*
This Fig 7 has been removed.

*Figure 8. Same as figure 7, we can hardly read the labels or identify the colours of the lines.*
This Fig 8 has been removed.

**References**
Blonquist, J., et al. (2009). "Evaluation of measurement accuracy and comparison of two new and three traditional net radiometers." Agricultural and Forest Meteorology **149**(10): 1709-1721.Gu,
 L., Meyers, T., Pallardy, S. G., Hanson, P. J., Yang, B., Heuer, M., . . . Wullschleger, S. D. (2006). Direct and indirect effects of atmospheric conditions and soil moisture on surface energy partitioning revealed by a prolonged drought at a temperate forest site. Journal of Geophysical Research: Atmospheres (1984–2012), 111(D16).
Twine, T. E., Kustas, W., Norman, J., Cook, D., Houser, P., Meyers, T., . . . Wesely, M. (2000). Correcting eddy-covariance flux underestimates over a grassland. Agricultural and Forest Meteorology, 103(3), 279-300.
Wilson, K., Goldstein, A., Falge, E., Aubinet, M., Baldocchi, D., Berbigier, P., . . . Field, C. (2002). Energy balance closure at FLUXNET sites. Agricultural and Forest Meteorology, 113(1), 223-243.

---

## Author Comment (AC2) · 26 May 2016

**General comments**

We thank Reviewer#2 for the positive revision of this manuscript, and for contributing to its improvement. According to his/her general comments, we rewrote the abstract and the conclusion to include more specific findings of our study. In-depth explanations were given as to EBR results that we found. The seasonal trend of the fluxes was shown in our study, including the highlight of years when data were poor for further analysis. We hope that this effort will improve the manuscript, by strengthening the weak points highlighted by this Reviewer. We tried to answer every comment in detail (see supplement file).

**Specific comments**

*Line 21-24: This paragraph is not really needed in the Abstract.*

Thank you for the comment. The abstract has been rewritten as:

L16-35: Flux towers provide essential terrestrial climate, water and radiation budget information needed for environmental monitoring and evaluation of climate change impacts on ecosystems and society in general. They are also intended for calibration and validation of satellite-based earth observation and monitoring efforts, such as assessment of evapotranspiration from land and vegetation surfaces using surface energy balance approaches.

In this paper, 15 years of Skukuza eddy covariance data, i.e. from 2000 to 2014, were analysed for surface energy balance closure and partitioning. The surface energy balance closure was evaluated using the ordinary least squares regression (OLS) of turbulent energy fluxes (sensible (H) and latent heat (LE)) against available energy (net radiation (Rn) less soil heat (G)). Partitioning of the surface energy during the wet and dry seasons was investigated, as well as how it is affected by atmospheric vapor pressure deficit (VPD), and net radiation.

After filtering years with bad data, our results show an overall mean surface energy balance closure of 0.93. Seasonal variations of EBR also showed summer had best EBR with winter having the least closure. Nocturnal surface energy closure was lowest, and this was linked to low friction velocity during night-time, and an increase in friction velocity showed an increase in closure. The high surface energy balance closure gives confidence on the usability of these data for calibrating and validating

The surface energy partitioning of this savanna ecosystem showed that sensible heat flux dominated the energy partitioning between March and October, followed by latent heat flux, and lastly the soil heat flux, except during the wet season where latent heat flux was larger than sensible heat flux. An increase in net radiation was characterised by an increase in both LE and H, with LE showing a higher rate of increase than H in the wet season, and the reverse happening during the dry season. An increase in VPD is characterised by a decrease in LE and increase in H during the wet season, and an increase of both fluxes during the dry season.

*Line 27: Please cancel the word concept.*

See above response.

*Line 39: Because there are besides canopy and ground heat storage other minor flux terms, I suggest to rewrite the sentence as follows: ": : : (G) heat fluxes and other minor flux terms such as heat stored by the canopy and ground."*

Line 40: The sentence has been modified, and now reads:"…and other minor fluxes such as heat stored by the canopy and the ground."

*Line 65: Please rewrite as follows: ": : : is an accepted performance criterion of EC flux data : : :".*
*Remark: Please use everywhere the introduced abbreviations.*

Line 64: Corrected.

*Line 86: The abbreviation EB was not introduced before.*

Line 82: It has been introduced as "energy balance".

*Table 1: Please state here the number of replicates of the soil heat flux measurements. And it remains unclear, whether the authors considered the soil heat storage change in the upper 5-cm layer in their calculation of the soil heat flux at the surface. Please explain!*

Thank you for your comment.

The information on the replicates of soil heat flux measurements has been included in Table 1 as "Soil heat flux at 5 cm depth with 3 replicates, i.e. two under tree canopies and one on open space". In this study, however, we did not consider the heat storage terms because we did not have the full dataset of soil temperature and soil moisture to estimate the soil heat storage. Other studies ignore the heat storage terms based on the assumption that they have opposing behaviours at day and night-time, hence cancelling their overall effect (Papale et al. (2006)). However, for the sake of reporting on sources of error in our study, it is important that we mention that neglecting the minor storage terms is in itself a source of error, and their inclusion could have improved the EBC.

*Table 1: Please state the exact installation depths of the CS615 probes.*

The information has been added on Table 1 – "Volumetric soil moisture content at 3, 7, 16, 30, and 50 cm in the clayey *Acacia* – dominated soils downhill of the tower, and 5, 13, 29, and 61 cm in the sandier *Combretum* – dominated soils uphill".

*Line 117: Please explain here or in the Discussion why you decided to switch from a closed-path to an open-path gas analyser. Is the change in the instrumentation somehow related to the performance of the EC system?*

This has been explained:

Line 195–197: The closed-path gas analyser was changed to open-path gas analyser in 2006. An analysis of the 2006 data (which had very low data completeness of 7.59 %) showed that there were no measurements recorded until September, possibly due to instrument failure. In their study, Wilson et al. (2002) mention that there were no differences in the EBR between sites using open and closed path gas analyzers.

*Line 120: As in line 117: Please explain why you did not continue to measure net radiation with the CNR2 sensor.*

The explanation has been given as:

L205-211: Between 2000 and 2004, the CNR2 net radiometer was used to measure long and shortwave radiation, and these were combined to derive Rn. However, when the pyrgeometer broke down in 2004, Rn was derived from measured shortwave radiation and modelled longwave radiation until the CNR2 was replaced by the NRLite net radiometer in 2009. This was a high source of error, as shown by the low EBR between 2004 and 2008. The closed-path gas analyser was also changed to open-path gas analyser in 2006. An analysis of the 2006 data (which had very low data completeness of 7.59 %) showed that there were no measurements recorded until September, possibly due to instrument failure.

*Line 157: The summation sign is not needed here. Write the equation simply as (H+LE)/(Rn-G).*
Corrected.

*Line 174: The authors should give at the beginning of the Results section an overview of the weather conditions over the last 15 years. Which years were particularly dry or wet. Which years were particularly warm or cold. Did you observe any long-term trend in the weather data.*
Line 172-186: Fig 1 shows the 15-year average daily temperature at the Skukuza flux tower. The annual average temperatures over the 15-year period ranged between 21.13°C in 2012 and 23.23 °C in 2003, with a 15-year average temperature of 22.9 °C. A slight decrease in temperature from 2000 to 2014 was observed. While 2003 was the hottest year, it was also the driest year, with annual rainfall of 273.6 mm, and 2002 also recording very low rainfall of 325.4 mm, both receiving rainfall amounts below the recorded mean annual. The wettest years were 2013, 2000, 2014 and 2004 which received 1414, 1115.6, 1010.2 and 1005.7 mm, respectively. 2007 and 2008 had incomplete rainfall data records to assess their annuals. The low rainfall during 2000-2003 seasons was also reported by Kutch et al. (2008), who were investigating the connection between water relations and carbon fluxes during the mentioned period.

*Line 176: From Figure 1 I would expect that the year 2013 was the year with the largest number of missing values and not the year 2001. Please explain.*
Line 190: This has been rectified.

*Line 214: What do you mean here with combined? Please explain. Did you not exclude where the years with low EBRs? Why does the mean EBR here does not agree with the figure (0.93) you gave in chapter 3.1.1?*

Here were took the whole 15-year data, excluding the years with bad data (2000-2004 and 2013) and partitioned them into seasons to determine the seasonal EBR.

*Line 236-242: Here it is not sufficient to explain the low EBR over the night time by referring to other studies. Please check you statement/conclusion against your own data. Are low EBRs related to low friction velocities?*

We did the analysis on how whether friction velocity is related to low EBR at night and found the results below:

Line 259-268: To understand the effect of friction velocity on the energy balance closure, years 2010 and 2012, which had friction velocity data, were used. Using friction velocity, the data were separated into 4 25-percentiles, and the EBR and OLS evaluated. Results show that the first quartile, the EBR was 3.94, with the 50-percentile at 0.99, the third quartile at unity, and the fourth quartile at 1.03 (Fig 5). The slopes were between 1.01 and 1.12, with the intercepts ranging between -9.26 and -0.17 Wm$^{-2}$, whereas R$^2$ were 0.82, 0.86, 0.85 and 0.81 for the first to the fourth quartiles, respectively.

 A quick assessment shows that the time associated with the low friction velocities, i.e. the first quartile are night-time data constituting 81 % of the whole first quartile dataset, and the last quartile had the highest number of daytime values at 79.29 % of the fourth quartile dataset.

**(Figure 1)**

*Line 249: I think it would be better to compile all the numbers in a table, and please do not aggregate the data to multi-year daily means. You lose so much information.*
*The authors should think about, for example, to give mean, minimum, and maximum monthly fluxes for every year.*

The annual means of the surface energy components were summarized as suggested in Table 2 as shown below:

**Table 1: Statistical summary of annual values of the energy balance components**

| Year | % data completion | | H | LE | G | Rn |
|------|------|------|------|------|------|------|
| **2000** | 14.16 | Max | 470.31 | 422.89 | 191.53 | 817.60 |
| | | Min | -139.77 | -72.43 | -61.60 | -95.93 |
| | | Mean | 45.82 | 36.11 | 5.32 | 91.46 |
| **2001** | 12.78 | Max | 790.82 | 513.09 | 292.87 | 899.90 |
| | | Min | -159.87 | -85.95 | -90.27 | -116.58 |
| | | Mean | 58.56 | 43.68 | 9.27 | 128.27 |
| **2002** | 17.77 | Max | 415.93 | 174.07 | 171.93 | 583.30 |

|      |       |      |         |         |         |         |
|------|-------|------|---------|---------|---------|---------|
|      |       | Min  | -117.66 | -89.16  | -86.00  | -122.21 |
|      |       | Mean | 61.35   | 10.29   | 4.10    | 90.72   |
| **2003** | 41.50 | Max  | 556.21  | 308.71  | 217.60  | 879.30  |
|      |       | Min  | -92.99  | -97.81  | -106.23 | -116.04 |
|      |       | Mean | 58.15   | 21.68   | 6.17    | 94.53   |
| **2004** | 28.21 | Max  | 505.36  | 498.10  | 129.96  | 925.30  |
|      |       | Min  | -150.08 | -89.07  | -69.76  | -5.88   |
|      |       | Mean | 56.46   | 17.99   | 7.97    | 156.10  |
| **2005** | 35.37 | Max  | 606.28  | 737.43  | 288.20  | 933.20  |
|      |       | Min  | -130.40 | -97.00  | -107.37 | -4.92   |
|      |       | Mean | 51.43   | 17.82   | 0.99    | 159.09  |
| **2006** | 7.59  | Max  | 583.66  | 331.25  | 335.30  | 1003.30 |
|      |       | Min  | -72.45  | -119.09 | -72.80  | -6.56   |
|      |       | Mean | 84.67   | 35.94   | 19.69   | 247.70  |
| **2007** | 48.77 | Max  | 552.93  | 426.34  | 340.67  | 1011.30 |
|      |       | Min  | -131.40 | -130.79 | -129.70 | -6.71   |
|      |       | Mean | 59.04   | 14.32   | 4.14    | 169.84  |
| **2008** | 54.30 | Max  | 616.43  | 439.76  | 238.57  | 1038.50 |
|      |       | Min  | -140.13 | -144.97 | -104.60 | -5.91   |
|      |       | Mean | 63.06   | 26.30   | 6.22    | 191.26  |
| **2009** | 42.69 | Max  | 551.34  | 776.62  | 328.93  | 1060.50 |
|      |       | Min  | -96.68  | -135.43 | -94.20  | -155.90 |
|      |       | Mean | 55.42   | 96.54   | 6.87    | 207.77  |
| **2010** | 57.65 | Max  | 626.68  | 624.38  | 199.33  | 888.00  |
|      |       | Min  | -173.11 | -135.62 | -66.35  | -180.70 |
|      |       | Mean | 57.23   | 52.54   | 3.74    | 105.10  |
| **2011** | 41.34 | Max  | 591.16  | 688.46  | 171.27  | 832.00  |
|      |       | Min  | -135.77 | -127.02 | -58.59  | -96.50  |
|      |       | Mean | 63.88   | 73.11   | 1.75    | 127.94  |
| **2012** | 27.62 | Max  | 572.11  | 566.88  | 185.80  | 899.00  |
|      |       | Min  | -171.83 | -148.49 | -50.92  | -99.69  |
|      |       | Mean | 59.25   | 52.49   | 2.16    | 111.31  |
| **2013** | 3.25  | Max  | 317.98  | 661.09  | 79.67   | 742.05  |
|      |       | Min  | -62.96  | -27.19  | -30.49  | -90.30  |
|      |       | Mean | 1.79    | 34.08   | -15.64  | -6.09   |
| **2014** | 28.66 | Max  | 533.46  | 726.31  | 89.50   | 893.00  |
|      |       | Min  | -238.65 | -134.39 | -33.36  | -89.70  |

| | | Mean | 59.37 | 69.55 | 1.18 | 147.30 |
|---|---|---|---|---|---|---|

*Line 255-260: Why did you limit this analysis to the year 2012? Please explain.*

With the inclusion of the description of the weather patterns for the 15 year period (Section 3.1), we then went on to do the analysis for the whole 15 years again. See Section 3.3.1.

*Line 270-272: Here the meaning of the months in brackets remains unclear. Also here it would be better to compile the data in a table. Please give in this context also the Bowen ratios.*

The months in brackets indicate when the values were recorded, for instance 97.48 Wm$^{-2}$ (June) means 97.48 Wm$^{-2}$ recorded in June.

*Figure 6: I suggest to plot the data as stacked columns and to include in the figure the residual! In this context it would be also important to give the possible range of fluxes due this residual (see e.g. Falge et al., 2005; Ingwersen et al., 2015) and to discuss whether the residual hampers the use of the data to validate satellite-based evapotranspiration methods.*

Thank you for your comment.

Given that the Skukuza flux tower EBR is already an average of 0.93, we conclude that these data can be used to validate satellite-derived evapotranspiration models. This would require applying methods to force closure before these data are used for validation, such as the Bowen ratio method. The scope of this paper does not cover the post-closure methods, as these will be part of a follow-up study.

*Line 338: This sentence reads strange. The sensible heat flux is not a part of net radiation. Please rewrite.*

The conclusion has been rewritten:

L357-377: This study investigated both surface energy balance and its partitioning into turbulent fluxes during the wet and dry seasons in a semi-arid savanna ecosystem in Skukuza using eddy covariance data from 2000 to 2014. The analysis revealed a mean multi-year energy balance ratio of 0.93, The variation of RBR based on season, time of day and as a function of friction velocity was explored. The seasonal EBR varied between 0.50 and 0.88, with winter recording the highest energy imbalance. Daytime EBR was as high as 0.72, with negative EBR for the nighttime. The high energy imbalance at night was explained as a result of stable conditions, which limit turbulence that is essential for the creation of eddies. The assessment of the effect of friction velocity on EBR showed that EBR increased with an increase in friction velocity, with low friction velocity experienced mainly during night-time.

The energy partition analysis revealed that sensible heat flux is the dominant portion of net radiation in this semi-arid region, except in summer, when there is rainfall. The results also show that

water availability and vegetation dynamics play a critical role in energy partitioning, whereby when it rains, vegetation growth occurs, leading to an increase in latent heat flux / evapotranspiration. Clearly an increase in Rn results in a rise in H and LE, however their increases are controlled by water availability. During the wet season, the rate of increase of LE is higher than that of H, whereas in the dry season the reverse is true. The rate of increase of LE is controlled by the availability of soil water (precipitation), and during the wet season it increases steadily with increasing Rn, whereas the rate of increase of H shows saturation with an increase in Rn. The opposite is true during the dry season, with limited water availability, the rate of increase of LE reaches saturation with increase in Rn and a steady increase of H with Rn increase. An increase in VPD, on the other hand, results in an increase in H and decrease in LE, with higher VPD experienced during the dry season, which explains the high H, although the evaporative demand is high.

*Line 331: Please revise the conclusions. They remain too general and on the level of text book knowledge. There must be something novel that we can learn from this 15-year long-term EC data record.*
The conclusion has been rewritten.

References: Falge, E., Reth, S., Brüggemann, N., Butterbach-Bahl, K., Goldberg, V., Oltchev, A., Schaaf, S., Spindler, G., Stiller, B., Queck, R., Köstner, B., and Bernhofer, C. (2005): Comparison of surface energy exchange models with eddy flux data in forest and grassland ecosystems of Germany, Ecol. Model., 188, 174–216.
Ingwersen, J., Imukova, K., Högy, P., Streck, T. (2015): On the use of the post-closure methods uncertainty band to evaluate the performance of land surface models against eddy covariance flux data, Biogeosciences, 12 (8), pp. 2311-2326.